# Effects of Cannabidiol Supplementation on Skeletal Muscle Regeneration after Intensive Resistance Training

**DOI:** 10.3390/nu13093028

**Published:** 2021-08-30

**Authors:** Eduard Isenmann, Sebastian Veit, Lynn Starke, Ulrich Flenker, Patrick Diel

**Affiliations:** 1Department of Molecular and Cellular Sports Medicine, Institute for Cardiovascular Research and Sports Medicine, German Sports University, 50933 Cologne, Germany; e.isenmann@dshs-koeln.de (E.I.); s.veit@dshs-koeln.de (S.V.); lynn@starke-team.de (L.S.); u.flenker@dshs-koeln.de (U.F.); 2Department of Fitness and Health, IST-University of Applied Sciences, 40233 Dusseldorf, Germany

**Keywords:** cannabidiol, CBD, recovery, muscle damage, performance, strength training, resistance training

## Abstract

Cannabidiol supplements (CBD) are increasingly consumed by athletes to improve regeneration. However, the evidence for the pro-regenerative effects of CBD in sports is quite limited. Therefore, our aim was to investigate the effects of a single CBD supplementation in a six-arm placebo-controlled crossover study after resistance training on performance and muscle damage. Before and after the resistance training, one-repetition maximum in the back squat (1RM BS), countermovement jump (CMJ), and blood serum concentrations of creatine kinase (CK) and myoglobin (Myo) were measured in healthy, well-trained participants. 16 out of 21 participants completed the study and were included in the analysis. In 1RM BS, a significant decrease was observed after 24 h (*p* < 0.01) but not after 48 and 72 h. A significant group difference was detected after 72 h (*p* < 0.05; *ES* = 0.371). In CMJ, no significant changes were observed. The CK and Myo concentrations increased significantly after 24 h (CK: *p* < 0.001; Myo: *p* < 0.01), 48 h (CK: *p* < 0.001; Myo: *p* < 0.01) and 72 h (CK: *p* < 0.001; Myo: *p* < 0.001). After 72 h, significant group differences were observed for both muscle damage biomarkers (CK: *p* < 0.05 *ES* = 0.24; Myo: *p* < 0.05; *ES* = 0.21). The results show small but significant effects on muscle damage and recovery of squat performance after 72 h. However, more data are required for clearer statements concerning potential pro-regenerative effects of CBD supplementation after resistance training.

## 1. Introduction

Intense training and performance result in damage and inflammation of the skeletal muscles [1,2,3]. Consequently, the performance of the trained skeletal muscles decreases after the intensive training sessions and needs time to recover [4,5,6]. In competitive sports, it is essential to minimise recovery time in order to provide further training stimuli as quick as possible and thus to increase performance. Dietary supplements are often used to this end [7,8,9]. Cannabidiol (CBD) products in particular are often consumed by elite athletes [10,11,12]. CBD is a non-psychoactive compound of *Cannabis sativa* [13]. Due to its structural differences to ∆^9^-tetrahydrocannabinol (∆^9^-THC), CBD has different affinities to cannabinoid receptors 1 and 2 [14,15,16,17]. Mechanistically, CBD can inhibit the NF-*κ*B signaling pathway and activate JAK/STAT kinase [18]. Therefore, CBD is used in a variety of diseases such as autoimmune encephalitis [19], rheumatoid arthritis [20], colitis [21], diabetes [22], and psoriasis [23], as well as Alzheimer’s [24], Parkinson’s [25], and Huntington’s Disease [26]. It has also anticonvulsant [14,27,28,29], antipsychotic [14,27,30], antiemetic [14,27], anticarcinogenic [31,32], and antidepressant [14,33] effects. Although regulations regarding its legality vary among countries worldwide [34,35], CBD was removed from the World Anti-Doping Agency (WADA) prohibited substances list in 2018 [36], based on evidence that CBD is safe and well-tolerated by humans [37,38,39]. While the effects and associated mechanisms of CBD in the human body have been studied [40,41,42], data in the context of sports and performance is still limited. Previous research has shown that roughly 27–42% of CBD users report improvement of sleep as reason for administration [43,44]. Improved sleep quality has been shown mainly in animal models [45,46] and not clearly in humans [47,48,49]. Additionally, first studies have shown a small effect on muscle soreness after CBD supplementation [50,51]. Concerning pro-regenerative effects of CBD in sport, currently only two studies exist which investigated biomarkers and performance parameters. In untrained males, the intake of CBD oil after six sets of ten maximal eccentric isokinetic elbow flexions did not result in any significant differences in perceived soreness, arm circumference, hanging joint angle, and peak torque between the groups [51]. In contrast, a pilot study [52] in highly trained men (>150% of body weight in the back squat [53]), showed an inhibitory effect of CBD on serum creatine kinase (CK) concentrations after three sets of 12 back squats at 70% 1RM plus three sets of 15 drop jumps landed in a deep squat. Consequently, based on the current lack of data, no definite conclusion can be drawn concerning the effects of CBD supplementation to promote regeneration processes after intensive strength training. Therefore, the aim of this study was to investigate the effects of a single CBD supplementation 24, 48, and 72 h after intensive strength training on muscle damage and performance. For this purpose, a placebo-controlled, double-blind, six-arm crossover study was conducted.

## 2. Materials and Methods

### 2.1. Participants

The study design was approved by the ethics committee of the German Sports University Cologne and was conducted in accordance with the Declaration of Helsinki. All participants were informed about the study design and confirmed their voluntary participation in written form. A total of 21 subjects were recruited for the study. People who were ill, injured, or were taking medication or dietary supplements were excluded. All subjects were healthy and experienced (at least one year) in strength training as demonstrated by back squat performances of >125% body mass [53]. A total of 16 subjects (Table 1) completed the study, while five subjects dropped out due to external reasons. To confirm the performance requirements, maximum strength was tested prior to the investigation. During the actual days of the study as well as 24 h before, no additional physical activities were allowed. Habitual diets were perpetuated.

### 2.2. Study Design

The study was carried out as a randomized, double-blind study in a six-arm crossover design and compared two conditions after an intensive strength training: the experimental condition using CBD (CBD) and the control condition using a placebo (PLA). Washout periods of two weeks, respectively, were placed in-between interventions. For neuromuscular measures, maximum strength and power were tested for the lower body. CK and Myoglobin (Myo) were used as biochemical measures. Pretests (T0) were performed for all four parameters prior to each intervention. Additionally, the same tests were executed again in three different sessions 24 (T24), 48 (T48), and 72 (T72) hours following training protocols. This results in three different PRE-POST interventions per condition for a total of six interventions (Figure 1). While the general order of conditions was the same, subjects were randomly assigned to start at one out of six conditions according to age, gender, and maximal strength.

### 2.3. Testing Protocol

For initial examinations, the subjects reported to the facility in the morning in rested and fasted state. Anthropometric data were recorded, and blood samples were taken (T0). Subsequently, the subjects consumed a standardized breakfast consisting of 100 g banana, 5 g honey, and 40 g or 60 g oatmeal (females or males, respectively). Thus, nutritional values accounted to about 270 kcal for female subjects and about 350 kcal for male subjects. After a 30-min break, warm-up started with five minutes of running and five minutes of dynamic stretching. After the warm-up, maximal power and strength were tested. This testing protocol was performed both PRE (T0) and POST training (T24, T48, T72). It was based on the guidelines of the National Strength and Conditioning Association (NSCA) for performance testing [54].

### 2.4. Skeletal Muscle Serum Creatine Kinase (CK) and Myoglobin (Myo)

CK and Myo serum concentrations were determined for all time points until subjects performed their POST intervention using the COBAS h 232 Point-of-Care-System (Roche Diagnostic Systems, Rotkreuz, Switzerland). Therefore, the sample size decreases with each POST intervention as values after the POST intervention would be comparable.

### 2.5. Strength and Power 1RM Back Squat (1RM) and Counter Movement Jump (CMJ)

Maximal strength and power performance tests were conducted at each PRE (T0) and POST (T24, T48, T72) intervention. To measure maximal power output, the subjects completed jump and reach tests (CMJ). Three jumps were averaged to estimate jump heights. Maximal strength was measured as the one-repetition maximum (1RM) in the back squat. Subjects performed four warm-up sets, starting with 10 repetitions at 50% of their estimated 1RM, followed by eight repetitions at 60–70%, four to six repetitions at 70–85%, and, finally, two to four repetitions at 80–95%. All loads were rounded to the nearest 2.5 kg step. Subjects rested for two minutes between warm-up sets. Before the first 1RM attempt, another three minutes of rest was taken. Subjects performed up to four attempts to establish the 1RM. The first attempts were taken at 90%, followed by a four-minute rest that was taken between all attempts.

### 2.6. Training Protocol

Following the T0 testing protocol, subjects performed a training protocol that intended to induce muscle damage. It included three sets of 12 back squats at an intensity of 70% of their 1RM with 150 s of rest between sets. In addition, subjects performed a combination of drop jumps from a 45 cm high box, landed in a deep squat, for three sets of 15 repetitions with 60 s of rest between sets. To be clear, this combination is a classic drop jump off the box where athletes are supposed to minimize contact time with the floor while still jumping as high as possible. This jump off the floor is then to be landed in a deep squat.

### 2.7. Supplementation

Following the training protocol, subjects drunk 60 mg CBD solubilisat with 250 mL water or a PLA drink directly after exercise. Specific CBD and PLA caps that fitted on a standard plastic bottle were produced that could only be distinguished by their color (red/white cap) (ATHENION GmbH, Berlin, Germany). There was no difference in taste or appearance between the two beverages. Other foods, dietary supplements, or caffeinated drinks were prohibited for the following three hours.

### 2.8. Statistical Analysis

Raw data were analyzed using the statistical software R version 4.0.4 (R Core Team, 2021). The raw data of CK and Myo were transformed to their natural logarithms (*ln*) before data analysis. The values of CMJ and of 1RM were used as-is. The individual time interval (∆t [h]) since the beginning of the investigation was introduced as an additional covariate. Data analysis was performed by means of linear mixed-effects models (LME). By definition, *ln*(CK), *ln*(Myo), CMJ, and 1RM represented the dependent variables. Model building was performed independently for each of these measures. We were precisely interested in the potential effects of CBD on the course of recovery. Therefore, all models invariably encompassed the interaction term of CBD with the recovery interval (RI, levels T0 through T72) as a fixed effect. Likewise, RI itself axiomatically was included as a fixed effect. Effect sizes (*ES*) of the fixed effects were calculated as *ES* = 2⋅*t*/*DF*,where *t* represents the *t*-test statistic and *DF* the corresponding degrees of freedom. Effect sizes were classified as trivial (*ES* < 0.2), small (0.2 < *ES* < 0.5), medium (0.5 < *ES* < 0.8), or large (*ES* > 0.8). Initially, random effects were merely assumed between the individual intercepts of each measure. Subsequently, ∆t was included as a random effect, where linear individual trends were assumed. The presence of potentially non-linear individual trends then was investigated by upgrading to 2^nd^ or 3^rd^ order natural splines of ∆t. Moreover, ∆t was invariably incorporated into the models as 1^st^ order continuous auto-regressive term, i.e., the data was assumed to be individually auto-correlated in time. After the development of an appropriate random effects structure, it was tested whether ∆t also contributes to general trends in the population, i.e., whether it represents a significant fixed effect. In either case, model comparisons were based on likelihood statistics and of changes of the Akaike information criterion (AIC).

## 3. Results

Means and standard errors of the mean (mean ± SEM) of all parameters for each group can be seen in Table 2. The means ± SEM for each parameter are depicted in Figure 2. Individual values can be found in Appendix A as well as Appendix A.

### 3.1. Physiological Adaptations over Time (Random Effects)

The models for *ln*(CK) and for *ln*(Myo) required 2^nd^ order natural splines of ∆t in order to appropriately describe individual trends. Both, the introduction of linear and 2^nd^ order terms resulted in significant improvements of the model fits (*p* < 0.01, respectively). The introduction of higher-order terms of ∆t corrupted the models as inferred by AIC increases. The individual trends of CMJ and 1RM, by contrast, were sufficiently described by linear terms. In both cases, the introduction yielded significant model improvements (*p* < 0.001, respectively). 2^nd^ order natural splines, subsequently, yielded no further improvements but undesirable increases in AIC. *ln*(CK) and *ln* (Myo) showed significant decreases during the course of the study (CK: *p* < 0.01, *ES* = −0.29; Myo: *p* < 0.05, *ES* = −0.25). Meanwhile, 1RM showed a significant increase (*p* < 0.05, *ES* = 0.34) (See Appendix A for the corresponding parameter estimates (∆t)).

### 3.2. Effects on Skeletal Muscle Damage

Appendix A show the fixed effects of the final models for *ln*(CK), *ln*(Myo), CMJ, and 1RM, respectively. In case of statistically significant effects (*p* < 0.05), the corresponding effect sizes (*ES*) have been added. In addition to the individual trends, *ln*(CK), *ln*(Myo), and 1RM required the introduction of a linear term for ∆t. *ln*(CK) was significantly elevated in both groups after 24 h (*p* < 0.001; *ES* = 0.69), 48 h (*p* < 0.001; *ES* = 0.53), and 72 h (*p* < 0.001; *ES* = 0.52). CBD attenuated the increase after 72 h, resulting in a significant group difference (*p* < 0.05; *ES* = −0.24). No significant effects of CBD on *ln*(CK) were observed at T24 and T48. Similarly, *ln*(Myo) showed a significant increase in both groups after 24 h (*p* < 0.01; *ES* = 0.30), 48 h (*p* < 0.01; *ES* = 0.28), and 72 h (*p* < 0.001; *ES* = 0.42). Again, after 72 h, CBD showed a dampening effect on *ln*(Myo) resulting in a significant group difference (*p* < 0.05; *ES* = 0.21).

### 3.3. Effects on Performance

CMJ did not exhibit significant changes at any time in both groups. It also did not change significantly over the course of the study (∆t). 1RM was significantly reduced in both groups after 24 h (*p* < 0.01, respectively) but not after 48 or 72 h. Although changes from T0 to T72 were not significant, CBD caused a significant group difference (*p* < 0.05; *ES* = 0.37). Changes in performance differed substantially between subjects. Therefore, it is worth looking at the individual changes to see how many subjects increased, decreased, or maintained in performance. After 24 h, five subjects of the CBD group and nine subjects of the PLA group showed decreases in their performance, while three subjects of each group improved. Eight subjects of the CBD group and four subjects of the PLA group were able to maintain their performance. After 48 h, five subjects of the CBD group and six subjects of the PLA group showed decreases, while three of the CBD group and six of the PLA group showed increasing performances. Eight and four subjects of the CBD and PLA group upheld performance, respectively. After 72 h, the CBD group featured two subjects with decreases in performance while the PLA group included eight. Seven subjects of the CBD group even increased their performance, while only three did so in the PLA group. Seven subjects of the CBD and four of the PLA group upheld performance, respectively.

## 4. Discussion

This study examined the effect of a single CBD supplementation following an intensive strength protocol on skeletal muscle damage and lower body performance. The results clearly show an increase in CK and Myo concentration in both groups at all time points (T24, T48, T72) with significant differences between the groups at T72 (*ES*: CK = −0.24, Myo = −0.21). The exercise induced muscle damage and the increase in CK and Myo concentration after T24 confirms observations from previous studies [1,2,3,52]. However, a single supplementation of CBD did not reduce muscle damage within 48 h. A longer time interval of at least 72 h, however, might be sufficient for effects of CBD to occur to detectable extend. Therefore, this study cannot confirm the inhibitory effect of CBD on CK concentrations after 24 h as observed in a pilot project with very well-trained strength athletes [52]. Furthermore, it can be assumed that different nutritional supplements can promote recovery at different rates despite similar mechanisms. Research has shown that a protein–carbohydrate combination can inhibit the increase in CK concentration after the same training protocol in athletes with similar performance levels [3]. One reason regarding the effects of CBD on muscle damage is possibly the absorption rate. Previous studies have shown that the maximum absorption of CBD can take between one to four hours [55]. Therefore, it can be speculated that post-exercise use may be too late to reduce muscle damage. Consequently, it may be beneficial to administrate CBD before training to conform the absorption to the end of training. Regarding performance outcomes, a significant decrease was observed in 1RM but not in CMJ. The results of 1RM are confirmed with observations of previous studies [1,3,52,56]. However, the decrease in performance in the CBD group is smaller than that in the pilot study [52]. This may be due to different performance levels of the subjects. The absolute strength values and training loads were significantly higher in the pilot study as compared to the present study (146.5 ± 21.6 vs. 118.6 ± 22.6 kg). This could explain the different changes in CK concentrations. One important observation of this study is the group difference between the CBD and PLA group at T72 (*p* < 0.05; *ES* = 0.37). In contrast to the PLA group, the CBD group was able to recover the squat performance and even to slightly increase it (Table 2). This observation is in line with the kinetics of CK and Myo concentrations. It seems that after 72 h, the CBD group has largely recovered and thus has restored its maximum performance. However, comparable effects cannot be observed for CMJ. Neither significant changes at any time, nor differences between groups were observed. It may be assumed that either the training protocol did not induce sufficient power reduction or that the Jump and Reach Test is not an ideal test for power performance. Consequently, jumping ability should ideally be measured with a force plate and, possibly, the loading protocol should be revised to measure power change in the CMJ. In addition to the influence of CBD supplementation on performance, the influence of measurement repetition was examined due to the six-arm study design. All models required the introduction of ∆t in a slightly differing manner. Fundamentally, *ln*(CK) and *ln*(Myo) showed significant negative population trends, while 1RM showed a linear increase. Although comparably small, effect sizes indicate the presence of adaptation processes throughout the study (*ln*(CK): −0.29; *ln*(Myo): −0.25; 1RM: 0.34. Cf. Appendix A). Physiologically, the findings are mutually consistent. However, the imperative precondition of significant experience in strength training was presumably not sufficiently met by all participants. Otherwise, the comparably small number of training bouts would not have induced detectable adaptations. Formally, individual intercepts or trends could also have been considered to constitute random effects nested within the respective arm of the study. This, however, would have presumed random scatter of the individual parameters between arms. After all, the order of the arms had been individually randomized to this end. This presumption, however, is false. Rather, ∆t must be considered to consistently take significant individual and possibly non-linear effects. For practical reasons, CBD and PLA were not administered in a strictly alternating order. Rather, the administration patterns were randomized independently within each of the subjects. Therefore, the factors ∆t, i.e., adaptation, and CBD confound fundamentally. Hence, the potential effects of CBD might be slightly obliterated. In fact, the first out of the six testing sequences merely featured six CBD but ten PLA treatments. But the highest proportions of CBD treatments precisely occurred during the midsection of the study (sequences 3 and 4, 9/16 and 10/16 CBD treatments, respectively) while the remaining sequences happened to be largely balanced. The potential effects of CBD are, thus, unlikely to become significantly biased by adaptation. One of the most advantageous features of LMEs is exactly to account for time-dependent covariates, if appropriately modeled. As is fundamentally desirable in sports science and sports medicine, adaption processes at individual and population levels have been reflected as far as possible here. The validity of the model estimates, therefore, appears largely untainted by this potential confounder. Although this study provides important findings concerning the effect of a single CBD supplementation after intensive strength training, there are also some limitations. The results of this study cannot be exactly compared to previous studies without mentioning differing strength levels. Due to the training restriction before and during the intervention, subjects were not able to follow their usual training routines. Very well trained strength athletes (squat performance above 150% of their BW) usually train more than four times a week to achieve appropriate training volumes for all muscle groups [57]. Therefore, it was impossible to recruit a population of highly trained athletes in this study design. In addition, the jump-and-reach test might not be sufficiently valid to monitor performance. Furthermore, no statements can currently be made about the effects of CBD after endurance training or after chronic supplementation and training. In order to minimise the limitations and to prevent possible adaptation processes induced by the interventions, future studies should focus on well to very well trained athletes [53,58] to assess the effect of CBD on recovery. In addition to performance and muscle damage parameters, inflammatory, immune, and antioxidative biomarkers should also be included in future studies for a more detailed analysis of the muscle healing process and adaption.

## 5. Conclusions

This study investigated the effects of CBD after 24, 48, and 72 h following a single resistance training in a placebo-controlled, six-arm crossover study. No strong effects were observed in biomarkers or performance parameters. However, small and significant effects of a single supplementation of CBD on CK and Myo concentrations were observed after 72 h. If CBD is supposed take stronger effects on recovery processes following intense strength training, continuous and repetitive supplementation will probably be required. For clearer statements, however, further studies on pro-regenerative and recovery effects of CBD after strength training or other sports are essential.

## Figures and Tables

**Figure 1 nutrients-13-03028-f001:**
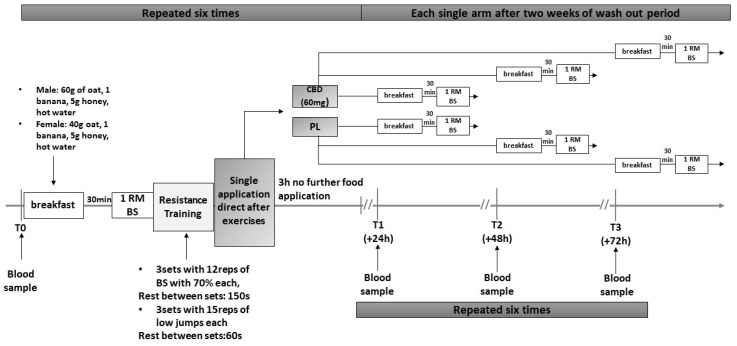
Study design. BS = Back Squat; reps = Repetitions; s = second; 1RM; One-Repetition Maximum; h = hours.

**Figure 2 nutrients-13-03028-f002:**
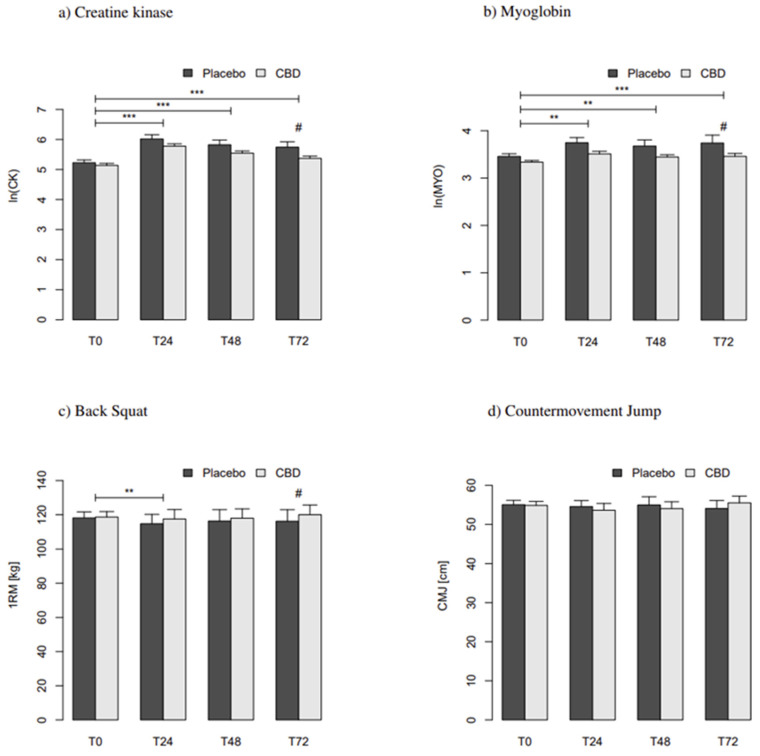
Barplots of the means ± SEM for the modeled parameters. (**a**) Skeletal Muscle Specific Creatine Kinase, (**b**) Myoglobin, (**c**) Back Squat, (**d**) Countermovement Jump. T0 through T24: Time, where figures indicate levels [h]. Asterisks indicate significant differences in time. *p* < 0.01, **; *p* < 0.001, ***. Hash indicates difference between groups at indicated level of time (#: *p* < 0.05).

**Table 1 nutrients-13-03028-t001:** Subjects information.

Parameter	$\bar{x}\pm s$
Age [Years]	24 ± 3
Height [cm]	181.4 ± 10.0
Body Mass [kg]	79.2 ± 13.7
CMJ [cm]	55.2 ± 8.5
1RM [kg]	118.5 ± 24.7
1RM/BM [kg]	1.4 ± 0.2

$\bar{x}\pm s$ = mean ± standard deviation; CMJ = Countermovement Jump; 1RM = One-Repetition Maximum; 1RM/BM = One-Repetition Maximum/Body Mass.

**Table 2 nutrients-13-03028-t002:** Empirical estimates of the muscle damage parameters (CK, Myo) and empirical estimates of performance parameters (1RM BS, CMJ) for given combinations of time and group levels.

		PLA (*n* = 16)			CBD (*n* = 16)			
Parameter	Units	PRE (T0)	POST	∆	PRE	POST	∆	ES
Creatine Kinase T 24	[U/L]	211.6 ± 16.9	399.9 ± 68.4	188.3 ± 58.4 ***	197.4 ± 19.1	351.5 ± 61.3	154.1 ± 61.1 ***	‒
Creatine Kinase T 48	[U/L]	196.8 ± 39.1	530.8 ± 237.5	333.9 ± 238.6 ***	188.6 ± 25.2	320.9 ± 65.9	132.4 ± 66.0 ***	‒
Creatine Kinase T 72	[U/L]	347.9 ± 151.3	3417.7 ± 2119.9	3069.7 ± 2106.6 ***	181.9 ± 25.2	232.6 ± 35.0	50.7 ± 44.7 ***/#	0.24
Myoglobin T 24	[ng/mL]	31.8 ± 8.3	38.4 ± 15.1	6.6 ± 9.2 **	31.9 ± 15.0	31.8 ± 8.3	−0.1 ± 12.7 **	‒
Myoglobin T 48	[ng/mL]	30.5 ± 10.0	33.9 ± 15.4	3.5 ± 19.0	28.3 ± 6.4	37.5 ± 25.6	9.2 ± 25.0 **	‒
Myoglobin T 72	[ng/mL]	30.3 ± 5.8	37.1 ± 15.8	6.8 ± 15.0 ***	28.1 ± 5.6	34.4 ± 11.3	6.3 ± 8.6 ***/#	0.21
Back Squat T 24	[kg]	118.0 ± 6.0	114.7 ± 5.5	−3.3 ± 1.5 **	118.1 ± 5.5	117.5 ± 5.5	−0.6 ± 0.8 **	‒
Back Squat T 48	[kg]	118.3 ± 6.0	116.3 ± 6.7	−2.0 ± 1.9	119.1 ± 5.6	118.0 ± 5.4	−1.1 ± 1.0	‒
Back Squat T 72	[kg]	118.2 ± 6.5	116.2 ± 6.8	−2.0 ± 1.6	118.8 ± 5.8	120.0 ± 5.6	1.3 ± 0.9 #	0.37
Counter Movement Jump T 24	[cm]	55.7 ± 2.0	54.6 ± 1.5	−1.2 ± 0.7	54.4 ± 1.8	53.6 ± 1.7	−0.8 ± 0.4	‒
Counter Movement Jump T 48	[cm]	54.9 ± 2.0	55.0 ± 2.1	0.1 ± 0.6	54.9 ± 1.8	54.0 ± 1.8	−0.8 ± 0.6	‒
Counter Movement Jump T 72	[cm]	54.4 ± 2.0	54.1 ± 2.1	−0.4 ± 0.8	55.3 ± 1.7	55.5 ± 1.7	0.2 ± 0.6	‒

Figures represent means ± SEM. PRE: pre training, POST: post training. ∆: corresponding differences. Asterisks indicate significant differences in time (*: *p* < 0.05, **: *p* < 0.01, ***: *p* < 0 .001). Hash indicates significant difference between groups (#: *p* < 0.05). Effect sizes (*ES* = 2·*t*/*DF*) are indicated in case of significant group effects.

## Data Availability

The data presented in this study are available in Appendix A.

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
