# Peer review of "Effects of Cannabidiol Supplementation on Skeletal Muscle Regeneration after Intensive Resistance Training"

_nutrients, 2021, doi:10.3390/nu13093028_

Round 1

Reviewer 1 Report

This study is interesting and contributes to the elucidation of the physiological effects of CBD. The manuscript is well described and statistical analysis is appropriate. However, there is something to be improved.

The manuscript would benefit tremendously from language editing by a professional editor. There are some errors throughout the manuscript. For example, “a significant decrease were” in lines 8-9 are incorrect. “IN CMJ” in line 10, “IN” should be described as “In”.

In the Abstract, the description of the intergroup between the placebo group and the CBD group is unclear. Write in an easy-to-understand manner using the word “placebo”. If the number of words exceeds 200, the authors can delete or abbreviate the sentences in lines 13-17 because the limitations of this study are well discussed in the Discussion section and are mentioned in the Conclusions section.

Author Response

Dear Reviewer 1,

thank you very much for your important comments. You will find the point by point answers in the attachement.

Reviewer 2 Report

The study by Isenmann E et al studied the effect of a single CBD supplementation after a resistance exercise session on performance and muscle damage. The authors reported small but significant effects of CBD supplementation on muscle damage and recovery of squat performance after 72 h. The study is well designed and the data is interesting.

A few comments for the authors:

  1. For 2.4 CK and Myo were determined in serum samples, should the title be changed to “Serum CK and Myo”?
  2. Mybe “ one application of CBD” should be replaced with “ one supplementation of CBD “
  3. The language presentation in the introduction and discussion could be improved, for example, Line 203 Please change “show clearly” to “clearly show”.

Author Response

Dear Reviewer 2,

thank you very much for your important comments. You will find the point by point answers in the attachement.
